# Clinical characteristics and disease course of splanchnic vein thrombosis in gastrointestinal cancers: A prospective cohort study

**Minsu Kang**[1], **Koung Jin Suh**[1], **Ji-Won Kim**[1], **Ja Min Byun**[2], **Jin Won Kim**[1], **Ji Yun Lee**[1], **Jeong-Ok Lee**[1], **Soo-Mee Bang**[1], **Yu Jung Kim**[1], **Se Hyun Kim**[1], **Jee Hyun Kim**[1], **Jong Seok Lee**[1], **Keun-Wook Lee**[1] *

1 Department of Internal Medicine, Seoul National University Bundang Hospital, Seoul National University College of Medicine, Seongnam, Republic of Korea, 2 Department of Internal Medicine, Seoul National University Hospital, Seoul National University College of Medicine, Seoul, Republic of Korea

* hmodoctor@snubh.org

## Abstract

**Data Availability Statement:** All relevant data are within the manuscript and its Supporting Information files.

## Purpose

Splanchnic vein thrombosis (SpVT) in solid tumors has not been well investigated. Therefore, the treatment guidelines for SpVT are not well established. We aimed to conduct this prospective study to investigate the clinical characteristics and risk factors influencing survival in patients with gastrointestinal cancer with SpVT.

## Materials and methods

Fifty-one patients with gastrointestinal cancer diagnosed with SpVT were prospectively enrolled. The clinical characteristics and courses of SpVT were analyzed.

## Results

SpVT occurred in various clinical situations (at the time of initial cancer diagnosis or tumor recurrence after curative therapy, in the postoperative period, during chemotherapy, or in the period of end-of-life care). Among the total patients, 90.2% had no SpVT-related symptoms at initial SpVT diagnosis, and 82.4% did not receive any anticoagulation therapy. The clinical course of SpVT during the follow-up varied: (1) spontaneous resorption without any anticoagulation (47.1%), (2) resorption with anticoagulation (3.9%), (3) persistent thrombosis without progression (17.6%), and (4) SpVT extension (31.4%). Although the SpVT showed extension in some cases, most of them did not cause symptoms or had little impact on the patient's cancer treatment course. During the follow-up period, 23 patients died, all of which were caused by tumor progression. In the multivariable analysis, performance status and clinical situation at the time of SpVT diagnosis were significant prognostic factors.

**Funding:** This research was partially funded by the Seoul National University Bundang Hospital Research Fund (Number: 02-2017-036). There was no additional external funding received for this study.

**Competing interests:** The authors have declared that no competing interests exist.

## Conclusions

Clinicians could adopt a strategy of close observation for incidentally detected SpVT in patients with gastrointestinal cancer. Anticoagulation should be considered only for SpVT cases selected strictly, weighing the risks and benefits.

## Introduction

The association between venous thromboembolism (VTE) and cancer is well recognized. However, compared with deep-vein thrombosis (DVT)/pulmonary embolism (PE), the clinical characteristics and prognostic impact of splanchnic vein thrombosis (SpVT) in solid tumors are not as well-studied. SpVT is a clinical manifestation of VTE that involves intra-abdominal veins such as the portal, mesenteric, hepatic, and splenic veins in an isolated site or in multiple sites simultaneously [1]. As data on the clinical course and prognostic impact of SpVT in patients with solid tumors are lacking, the treatment guidelines for SpVT are not well established, particularly for incidentally detected SpVT cases [2,3].

Previous studies have compared cancer patients diagnosed with and those without SpVT to elucidate the prognostic impact of SpVT. Søgaard *et al.* [4] described SpVT as a prognostic factor for short-term survival in patients diagnosed with liver or pancreatic cancer. In Afzal *et al.*'s study [5], SpVT was associated with worse survival in patients with advanced pancreatic adenocarcinoma, and anticoagulant therapy for SpVT did not affect the increased mortality. By contrast, Choi *et al.* [6] and Lee *et al.* [7] analyzed the different characteristics and prognostic impact of DVT/PE and SpVT in patients with colorectal cancer and those with gastric cancer, respectively. Both studies concluded that only DVT/PE has a negative effect on survival, whereas SpVT has no prognostic significance. Most reports on SpVT have been conducted based on retrospective analysis of cancer patient cohorts.

To our knowledge, no prospective studies have described the clinical features of SpVT in patients with solid tumors. Therefore, we aimed to conduct this prospective study to describe the clinical characteristics and identify the risk factors influencing survival in patients with gastrointestinal cancer with SpVT.

## Materials and methods

This prospective observational cohort study was conducted in a single tertiary teaching hospital [Seoul National University Bundang Hospital (SNUBH)] in Korea. Patients with gastrointestinal cancer who visited the oncology clinic of SNUBH were eligible to enter the study cohort. Inclusion criteria were as follows: (1) age ≥18 years; (2) pathologically confirmed gastrointestinal cancer (gastric, colorectal, and small intestinal cancers); and (3) diagnosis of SpVT by either abdominal computed tomography (CT) or magnetic resonance imaging (MRI). The patients who met the inclusion criteria and agreed to participate in this study were consecutively enrolled between June 2017 and July 2020. Since it was planned to enroll as many SpVT patients as possible during the study period, sample size and power calculations were not performed at the start of this study. Abdominal imaging examination was performed as a usual practice to diagnose the cancer, to evaluate the treatment effect, to identify the cause of abnormal symptoms, or to check for recurrence after completion of treatment, and it was not additionally performed for this study. Changes in SpVT were assessed by the abdominal imaging studies (CT or MRI) during the follow-up and the imaging examination interval was

determined according to the clinical judgment of the attending physicians. Non-Korean patients were excluded from the study cohort. Written informed consent was obtained from all patients, and the Institutional Review Board of SNUBH approved this study.

The following data were collected for this study: (1) demographics such as age, sex, height, weight, comorbid disease, and Eastern Cooperative Oncology Group performance status (ECOG PS); (2) data related to gastrointestinal cancer such as diagnosis code, date of diagnosis, laboratory blood test results, abdominal CT or MRI, surgery, chemotherapy, and radiation therapy; (3) data related to SpVT such as location, date of identification, initial symptoms and signs (abdominal pain, hepatomegaly, ascites, etc.), treatment, change of SpVT-related symptoms, recurrent thrombosis (including extension of pre-existing SpVT or new SpVT) during follow-up, and bleeding events after anticoagulation therapy; and (4) survival data, date of death, and reason for death in cases of death.

The outcomes of interest in this observational study were clinical characteristics and course of SpVT including anticoagulation therapy, recurrent thrombosis and bleeding events during follow-up, and risk factors influencing survival in patients with SpVT. Data were analyzed using IBM SPSS Statistics (IBM Corp., Armonk, NY, USA). The Kaplan–Meier method was used to calculate the overall survival (OS). OS was defined as the time period between the date of SpVT diagnosis and the date of death due to any cause or censoring. In univariable analyses, log-rank tests were performed to examine differences in survival outcomes among the comparison groups. Age, sex, tumor-related characteristics (primary tumor, pathology and stage), parameters related to the patient's status at the time of diagnosis of SpVT (ECOG PS and laboratory blood test results (hemoglobin level, white blood cell count, platelet count and albumin level), and parameters related to SpVT (location, the presence of related symptoms and the clinical situation at the time of SpVT diagnosis) were included in the univariable analyses. Cox proportional hazards models were used to analyze the influence of specified risk factors on survival outcomes in the multivariable analysis. Variables with $P$-value $< 0.05$ in the univariable analysis were included using the 'enter' method in the multivariable Cox proportional hazards model analysis. If there was high intercorrelation between the selected variables, one variable was selected and included in the multivariable analysis model, and it was checked whether the analysis result was the same when the other variable was selected instead of the first selected one. $P$-values less than 0.05 were considered significant.

## Results

### Patient characteristics

A total of 51 patients who met the eligibility criteria were consecutively enrolled (Table 1). Patients had gastric cancer [n = 25: adenocarcinoma (n = 24) and neuroendocrine carcinoma (n = 1)], colorectal cancer (n = 24: adenocarcinoma only), duodenal cancer (n = 1), and gastrointestinal stromal tumor (n = 1). The median age was 61 years (range, 31–82 years), and 76.5% were men. The distribution of tumor stages was as follows: stage II (11.8%), stage III (25.5%), and stage IV (62.7%). Prior major abdominal surgery, including both curative and palliative operations, was performed in 41 patients (80.4%). Previous radiotherapy and chemotherapy [palliative or peri-operative (prior to or following surgery)] were conducted in 23.5% and 74.5% of the patients, respectively. As of December 2, 2020 (the data cutoff date for analysis), the median follow-up period of all patients was 19.9 months (range, 0.3–41.0 months) and the median duration of follow-up of survivors was 27.7 months (range, 3.6–41.0 months). One patient immigrated to a foreign country (follow-up duration; 6.6 months), so further survival follow-up could not be done.

**Table 1. Patient characteristics.**

| | Total n = 51 | Gastric cancer n = 25 | Colorectal cancer n = 24 | Other cancer[a] n = 2 |
|---|---|---|---|---|
| Number of patients, n (%) | 51 (100.0%) | 25 (100.0%) | 24 (100.0%) | 2 (100.0%) |
| Age, n (%) (median, range) | 61 (31–82) | 58 (43–82) | 62 (31–78) | 54 (47–61) |
| < 70 | 41 (80.4%) | 20 (80.0%) | 19 (79.2%) | 2 (100.0%) |
| ≥ 70 | 10 (19.6%) | 5 (20.0%) | 5 (20.8%) | - |
| Sex, n (%) | | | | |
| Male | 39 (76.5%) | 19 (76.0%) | 19 (79.2%) | 1 (50.0%) |
| Female | 12 (23.5%) | 6 (24.0%) | 5 (20.8%) | 1 (50.0%) |
| BMI (median, range) | 22.3 (15.5–31.6) | 21.2 (15.5–28.0) | 23.2 (19.5–31.2) | 28.3 (24.9–31.6) |
| Smoking[b] | | | | |
| Never smoker | 25 (49.0%) | 13 (52.0%) | 11 (45.8%) | 1 (50.0%) |
| Former smoker | 18 (35.3%) | 8 (32.0%) | 9 (37.5%) | 1 (50.0%) |
| Current smoker | 7 (13.7%) | 4 (16.0%) | 3 (12.5%) | - |
| ECOG PS, n (%) | | | | |
| 0 | 15 (29.4%) | 6 (24.0%) | 8 (33.3%) | 1 (50.0%) |
| 1 | 30 (58.8%) | 13 (52.0%) | 16 (66.7%) | 1 (50.0%) |
| ≥ 2 | 6 (11.8%) | 6 (24.0%) | - | - |
| No. of comorbidities, n (%)[c] | | | | |
| 0 | 22 (43.1%) | 10 (40.0%) | 11 (45.8%) | 1 (50.0%) |
| 1 | 18 (35.3%) | 9 (36.0%) | 8 (33.3%) | 1 (50.0%) |
| 2 | 5 (9.8%) | 3 (12.0%) | 2 (8.3%) | - |
| ≥ 3 | 6 (11.8%) | 3 (12.0%) | 3 (12.5%) | - |
| Histologic group, n (%)[d] | | | | |
| WDAC | 3 (5.9%) | 1 (4.0%) | 1 (4.2%) | 1 (50.0%) |
| MDAC | 27 (52.9%) | 7 (28.0%) | 20 (83.3%) | - |
| PDAC | 13 (25.5%) | 11 (44.0%) | 2 (8.3%) | - |
| PCC | 5 (9.8%) | 4 (16.0%) | 1 (4.2%) | - |
| Mucinous adenocarcinoma | 1 (2.0%) | 1 (4.0%) | - | - |
| Others | 2 (3.9%) | 1 (4.0%) | - | 1 (50.0%) |
| Tumor stage, n (%) | | | | |
| II | 6 (11.8%) | 3 (12.0%) | 3 (12.5%) | - |
| III | 13 (25.5%) | 4 (16.0%) | 8 (33.3%) | 1 (50.0%) |
| IV | 32 (62.7%) | 18 (72.0%) | 13 (54.2%) | 1 (50.0%) |
| No. of metastatic organs, n (%) | | | | |
| 0 | 19 (37.3%) | 7 (28.0%) | 11 (45.8%) | 1 (50.0%) |
| 1 | 10 (19.6%) | 7 (28.0%) | 2 (8.3%) | 1 (50.0%) |
| 2 | 17 (33.3%) | 9 (36.0%) | 8 (33.3%) | - |
| ≥ 3 | 5 (9.8%) | 2 (8.0%) | 3 (12.5%) | - |
| Previous thromboembolic event | | | | |
| No | 49 (96.1%) | 23 (92.0%) | 24 (100.0%) | 2 (100.0%) |
| Yes | 2 (3.9%) | 2 (8.0%) | - | - |
| Major surgery | | | | |
| No | 10 (19.6%) | 7 (28.0%) | 3 (12.5%) | - |
| Yes (>3 months) | 27 (52.9%) | 14 (56.0%) | 13 (54.2%) | - |
| Yes (≤3 months[e]) | 14 (27.5%) | 4 (16.0%) | 8 (33.3%) | 2 (100.0%) |
| Radiotherapy | | | | |
| No | 39 (76.5%) | 20 (80.0%) | 18 (75.0%) | 1 (50.0%) |
| Yes (>3 months) | 10 (19.6%) | 5 (20.0%) | 4 (16.7%) | 1 (50.0%) |

*(Continued)*

**Table 1.** (Continued)

| | Total n = 51 | Gastric cancer n = 25 | Colorectal cancer n = 24 | Other cancer[a] n = 2 |
|---|---|---|---|---|
| **Yes (≤3 months[e])** | 2 (3.9%) | 0 (0.0%) | 2 (8.3%) | - |
| **Chemotherapy** | | | | |
| **No** | 13 (25.5%) | 6 (24.0%) | 7 (29.2%) | - |
| **Yes (>3 months)** | 6 (11.8%) | 1 (4.0%) | 4 (16.7%) | 1 (50.0%) |
| **Yes (≤3 months[e])** | 32 (62.7%) | 18 (72.0%) | 13 (54.2%) | 1 (50.0%) |

[a] Other cancer: duodenal cancer (n = 1) and gastrointestinal stromal tumor (n = 1).

[b] One patient did not report smoking history.

[c] No patient had thrombophilic disorders.

[d] WDAC, well-differentiated adenocarcinoma; MDAC, moderately differentiated adenocarcinoma; PDAC, poorly differentiated adenocarcinoma; PCC, poorly cohesive carcinoma; Others (n = 2) include large cell neuroendocrine carcinoma (n = 1) and gastrointestinal stromal tumor (n = 1).

[e] Within 3 months prior to diagnosis of SpVT.

## Clinical characteristics of SpVT

Table 2 shows the clinical characteristics of SpVT. The anatomical sites of SpVT development were the portal vein (66.7%), superior mesenteric vein (15.7%), inferior mesenteric vein (3.9%), others (5.9%; gastric vein (2.0%), right hepatic vein (2.0%), and internal iliac vein (2.0%)), and multiple sites (7.8%). In gastric cancer, inferior mesenteric venous thrombosis was not observed.

**Table 2. Clinical characteristics of splanchnic venous thrombosis (SpVT).**

| | Total n = 51 | Gastric cancer[a] n = 25 | Colorectal cancer n = 24 | Other cancer[b] n = 2 |
|---|---|---|---|---|
| **Location of SpVT (at the time of initial diagnosis of SpVT)** | | | | |
| Portal vein | 34 (66.7%) | 18 (72.0%) | 14 (58.3%) | 2 (100.0%) |
| Superior mesenteric vein | 8 (15.7%) | 4 (16.0%) | 4 (16.7%) | - |
| Inferior mesenteric vein | 2 (3.9%) | - | 2 (8.3%) | - |
| Others | 3 (5.9%) | 1 (4.0%)[c] | 2 (8.3%)[d] | - |
| Multiple sites | 4 (7.8%) | 2 (8.0%)[e] | 2 (8.3%)[f] | - |
| **Clinical situation (at the time of initial diagnosis of SpVT)** | | | | |
| Diagnosis of cancer (initial diagnosis) | 8 (15.7%) | 4 (16.0%) | 4 (16.7%) | - |
| Tumor recurrence (after curative therapy) | 4 (7.8%) | 3 (12.0%) | 1 (4.2%) | - |
| After surgery | 14 (27.5%) | 4 (16.0%) | 9 (37.5%) | 1 (50.0%) |
| During chemotherapy (without tumor progression) | 11 (21.6%) | 5 (20.0%) | 5 (20.8%) | 1 (50.0%) |
| During chemotherapy (with tumor progression) | 10 (19.6%) | 6 (24.0%) | 4 (16.7%) | - |
| Terminal phase (no more chemotherapy) | 4 (7.8%) | 3 (12.0%) | 1 (4.2%) | - |
| **SpVT-related symptoms (at the time of initial diagnosis of SpVT)** | | | | |
| Absent | 46 (90.2%) | 22 (88.0%) | 22 (91.7%) | 2 (100.0%) |
| Present | 5 (9.8%) | 3 (12.0%) | 2 (8.3%) | - |
| **Development of new SpVT-related symptoms during the follow-up period** | | | | |
| No | 49 (96.1%) | 24 (96.0%) | 23 (95.8%) | 2 (100.0%) |
| Yes | 2 (3.9%) | 1 (4.0%)[g] | 1 (4.2%) | - |
| **DVT or PTE (at the time of initial diagnosis of SpVT)** | | | | |
| Absent | 50 (98.0%) | 24 (96.0%) | 24 (100.0%) | 2 (100.0%) |
| Present | 1 (2.0%) | 1 (4.0%) | - | - |
| **Development of new DVT or PTE during the follow-up period** | | | | |

*(Continued)*

**Table 2.** (Continued)

| | Total n = 51 | Gastric cancer[a] n = 25 | Colorectal cancer n = 24 | Other cancer[b] n = 2 |
|---|---|---|---|---|
| No | 49 (96.1%) | 24 (96.0%) | 23 (95.8%) | 2 (100.0%) |
| Yes | 2 (3.9%) | 1 (4.0%)[h] | 1 (4.2%)[i] | - |
| **Clinical course of SpVT** | | | | |
| Spontaneous resorption without anticoagulation[j] | 24 (47.1%) | 8 (32.0%) | 14 (58.3%) | 2 (100.0%) |
| Resorption with anticoagulation[k] | 2 (3.9%) | 1 (4.0%) | 1 (4.2%) | - |
| Persistent thrombosis without progression | 9 (17.6%) | 6 (24.0%) | 3 (12.5%) | - |
| Extension of SpVT within the same vein | 12 (23.5%) | 7 (28.0%) | 5 (20.8%) | - |
| Extension to adjacent other veins beyond the existing location or new SpVT occurrence | 4 (7.8%) | 3 (12.0%) | 1 (4.2%) | - |
| **Anticoagulation treatment of SpVT** | | | | |
| No | 42 (82.4%) | 19 (76.0%) | 21 (87.5%) | 2 (100.0%) |
| Yes (at the time of initial diagnosis of SpVT) | 4 (7.8%) | 3 (12.0%) | 1 (4.2%) | - |
| Yes (at the time of aggravation of SpVT[l]) | 5 (9.8%) | 3 (12.0%) | 2 (8.3%) | - |

[a] One patient (M/64) had gastric adenocarcinoma and esophageal squamous cell carcinoma at the same time.

[b] Other cancer included duodenal cancer (n = 1) and gastrointestinal stromal tumor (n = 1).

[c] One patient (M/66) had SpVT in the gastric vein. During the follow-up, the SpVT in the gastric vein was extended to portal vein, superior mesenteric vein, and splenic vein thrombosis.

[d] One patient (F/56) had SpVT in the right hepatic vein and the other (M/70) had SpVT in the internal iliac vein.

[e] Among patients with gastric cancer, 2 patients had SpVT in multiple sites: one patient (F/58) had extensive thrombosis, which was located in the portal vein, superior mesenteric vein, inferior vena cava, and both common femoral veins; the other patient (M/59) had thrombus, which was located from the left gastric vein to the main portal vein.

[f] Among patients with colorectal cancer, 2 patients had SpVT in multiple sites: one patient (M/66) had SpVT in both the inferior mesenteric and portal veins; the other (M/56) had SpVT in the inferior mesenteric and splenic veins.

[g] One patient (M/58) with stage IV gastric cancer developed asymptomatic portal vein thrombosis during palliative chemotherapy. However, during the chemotherapy, severe abdominal pain and ileus developed and new superior mesenteric vein thrombosis was detected. In this case, mesenteric ischemia was strongly suspected and improved after use of dalteparin.

[h] Thrombosis in the confluent portion of the left internal jugular and subclavian veins due to left supraclavicular node metastasis.

[i] Simultaneous PTE and right common iliac vein tumor thrombus.

[j] Rates of spontaneous recanalization according to location of SpVT were as follows: portal vein 47.1% (16/34); superior mesenteric vein 50.0% (4/8); inferior mesenteric vein 50.0% (1/2); others 33.3% (1/3); and multiple sites 50.0% (2/4).

[k] Among these two patients who showed resorption after anticoagulation, one (M/63) had gastric cancer (pT3N1M0; stage IIB) and underwent total gastrectomy, distal pancreatectomy, and splenectomy. Postoperative focal thrombosis in the superior mesenteric vein was observed, and the thrombosis disappeared after anticoagulation (enoxaparin followed by warfarin). The other patient (M/65) had rectal cancer (clinical stage III) and developed portal venous thrombi in the right anterior and posterior segmental portal branches after ultralow anterior resection. Rivaroxaban was used, and the portal thrombosis disappeared.

[l] The SpVT extended from the existing location in 2 patients with gastric cancer. In 3 patients (one with gastric cancer and two with colorectal cancer), development of new SpVT was observed.

Abbreviations: SpVT, splanchnic vein thrombosis; DVT, deep vein thrombosis; PTE, pulmonary thromboembolism.

SpVT occurred in various clinical situations: (1) at the time of initial cancer diagnosis (15.7%); (2) at the time of tumor recurrence after curative therapy (7.8%); (3) postoperative period (27.5%); (4) during chemotherapy without evidence of tumor progression (21.6%); (5) during chemotherapy with the tumor progression [no clinical benefit from chemotherapy (19.6%)]; and (6) in the period of end-of-life care [terminal phase without further chemotherapy (7.8%)].

The majority of patients (90.2%) had no SpVT-related symptoms at the time of initial diagnosis of SpVT, and most patients (96.1%) did not develop new SpVT-related symptoms during

the follow-up period. One patient (2.0%) had DVT at the time of the initial diagnosis of SpVT. Development of new DVT or PTE during the follow-up period was observed in two patients.

Among the 51 patients, 42 (82.4%) did not receive any anticoagulation therapy for SpVT, while 9 (17.6%) received either low molecular weight heparin (n = 6), warfarin (n = 1), or direct oral anticoagulant (n = 2). In the 9 patients who were treated with anticoagulation, the median treatment duration was 4.7 months (range, 0.2–6.4 months). Rates of recanalization of SpVT were 57.1% (24/42) in non-anticoagulation group and 22.2% (2/9) in anticoagulation group.

Among 9 patients who received anticoagulation, four patients received anticoagulation treatment at the time of initial SpVT diagnosis: in 2 patients, SpVT was incidentally detected after surgery and anticoagulation was performed considering the possibility of worsening in the postop period; in the other two cases, SpVT was identified with tumor progression and anticoagulation was conducted for symptom management (in the first patient, ascites accompanying portal vein thrombosis was observed, and in the other case, DVT was detected simultaneously with SpVT). In five other patients, anticoagulation therapy was initiated at the time of SpVT aggravation (extension of pre-existing SpVT or development of new SpVT). Of these 9 patients who were treated with anticoagulation, 2 gastric cancer patients had bleeding events during the anticoagulation therapy. One patient was diagnosed with SpVT (multifocal thrombus in the portal veins) and a pseudoaneurysm in right proximal hepatic artery adjacent to metastatic tumor was also detected at the same time. During dalteparin administration, massive bleeding from the pseudoaneurysm developed and the patient died of bleeding. The other patient was diagnosed with SpVT with disease progression, and bleeding from primary gastric cancer occurred during dalteparin administration. After posterior gastric artery embolization, the patient continued the chemotherapy.

The clinical course of SpVT during the follow-up varied: (1) spontaneous resorption without any anticoagulation in 24 patients (47.1%); (2) resorption with anticoagulation in 2 (3.9%); (3) persistent thrombosis without progression in 9 (17.6%); (4) extension of SpVT within the same vein in 12 (23.5%); and (5) extension to other adjacent veins beyond the existing location or new SpVT occurrence in 4 (7.8%). Rates of spontaneous recanalization according to location of SpVT development were as follows: portal vein 47.1% (16/34); superior mesenteric vein 50.0% (4/8); inferior mesenteric vein 50.0% (1/2); others 33.3% (1/3); and multiple sites 50.0% (2/4). More detailed information on the clinical course of SpVT during the follow-up is shown in Table 2.

## Impact of SpVT on survival

Survival analyses were performed, and the results are presented in Table 3. The median OS and three-year OS rates after the diagnosis of SpVT were 29.1 months and 46.4%, respectively (S1 Fig). At the time of data cutoff, 27 subjects were alive; of these 27, three were transferred to a hospice for end-of-life care. One subject was lost to follow-up. Twenty-three patients died, caused by tumor progression in all cases; no patients died due to SpVT.

Additional analyses on prognostic factors related to survival in patients with SpVT were performed (Table 3). In univariable analyses, age $\geq$ 70 years (versus < 70 years), ECOG PS 2 [versus PS 0 or 1; Fig 1(A)], high tumor stage (IV versus II/III; S2 Fig), SpVT-related symptoms (present versus absent), and clinical situation at the time of diagnosis of SpVT [terminal phase without further chemotherapy/during chemotherapy (with tumor progression) versus initial diagnosis of cancer/tumor recurrence versus during chemotherapy (without tumor progression)/postoperative period; Fig 1(B)] were associated with shorter OS after the diagnosis of SpVT ($P < 0.05$). However, no significant difference was found in OS between patients with

**Table 3. Univariable and multivariable analyses on prognostic factors in patients with SpVT.**

| | n | Median (months) | 3-year OS rate | P | HR | 95% CI | P |
|---|---|---|---|---|---|---|---|
| **Sex** | | | | 0.920 | | | |
| Male | 39 | 27.1 | 43.0% | | - | - | - |
| Female | 12 | NR | 56.3% | | - | - | - |
| **Age** | | | | 0.011 | | | |
| < 70 years | 41 | NR | 56.6% | | 1.00 | - | - |
| ≥ 70 years | 10 | 13.2 | 0.0% | | 2.09 | 0.67–6.51 | 0.203 |
| **ECOG performance status** | | | | <0.001 | | | 0.006 |
| 0 | 15 | 29.1 | 49.2% | | 1.00 | - | - |
| 1 | 30 | NR | 54.0% | | 0.51 | 0.17–1.52 | 0.228 |
| ≥ 2 | 6 | 1.6 | 0.0% | | 9.01 | 1.06–76.61 | 0.044 |
| **Primary tumor** | | | | 0.231 | | | |
| Gastric cancer | 25 | 20.2 | 42.1% | | - | - | - |
| Colorectal cancer | 24 | 29.1 | 44.6% | | - | - | - |
| Others | 2 | NR | 100.0% | | - | - | - |
| **Tumor pathology** | | | | 0.189 | | | |
| WDAC/MDAC | 30 | NR | 57.1% | | - | - | - |
| PDAC | 18 | 20.9 | 33.9% | | - | - | - |
| Others | 3 | 20.2 | 33.3% | | - | - | - |
| **Stage** | | | | <0.001 | | | |
| II/III | 19 | NR | 83.0% | | - | - | - |
| IV | 32 | 13.6 | 23.7% | | - | - | - |
| **Location of SpVT** | | | | 0.979 | | | |
| Portal vein | 34 | NR | 51.8% | | - | - | - |
| Mesenteric vein (superior or inferior) | 10 | 27.1 | 45.7% | | - | - | - |
| Others | 3 | 29.1 | 33.3% | | - | - | - |
| Multiple sites | 4 | 23.0 | 50.0% | | - | - | - |
| **SpVT-related symptoms** | | | | 0.003 | | | |
| Absent | 46 | 29.1 | 50.3% | | 1.00 | - | - |
| Present | 5 | 5.3 | 20.0% | | 1.92 | 0.41–9.03 | 0.411 |
| **Clinical situation at the diagnosis of SpVT** | | | | <0.001 | | | 0.003 |
| After surgery | 14 | NR | 78.8% | | 1.00 | - | - |
| Initial diagnosis of cancer or tumor recurrence (after curative therapy) | 12 | 18.5 | 22.2% | | 5.57 | 1.00–33.46 | 0.051 |
| During chemotherapy (without tumor progression) | 11 | NR | 87.5% | | 0.80 | 0.72–8.91 | 0.857 |
| During chemotherapy (with tumor progression) | 10 | 5.3 | 15.2% | | 14.07 | 2.36–83.82 | 0.004 |
| Terminal phase (no more chemotherapy) | 4 | 1.2 | 0.0% | | 33.32 | 4.41–251.93 | 0.001 |
| **Albumin level (serum)** | | | | 0.180 | | | |
| ≥ 3.0 g/dL | 41 | 29.1 | 46.0% | | - | - | - |
| < 3.0 g/dL | 10 | 7.4 | 40.0% | | - | - | - |
| **Hemoglobin level (plasma)** | | | | 0.052 | | | |
| ≥ 10.0 g/dL | 34 | NR | 53.0% | | - | - | - |
| < 10.0 g/dL | 17 | 18.5 | 33.1% | | - | - | - |
| **White blood cell count level (plasma)** | | | | 0.395 | | | |
| ≥ 4000/μL | 40 | NR | 55.4% | | - | - | - |
| < 4000/μL | 11 | 23.0 | 15.7% | | - | - | - |
| **Platelet count level (plasma)** | | | | 0.531 | | | |
| ≥ 13,000/μL | 42 | 29.1 | 48.6% | | - | - | - |

(*Continued*)

**Table 3.** (Continued)

| | n | Median (months) | 3-year OS rate | P | HR | 95% CI | P |
|---|---|---|---|---|---|---|---|
| < 13,000/μL | 9 | 23.0 | 40.0% | | - | - | - |

Abbreviations: SpVT, splanchnic vein thrombosis; ECOG, Eastern Cooperative Oncology group; WDAC, well differentiated adenocarcinoma; MDAC, moderately differentiated adenocarcinoma; PDAC, poorly differentiated adenocarcinoma.

gastric cancer and those with colorectal cancer [Fig 1(C)] or among different SpVT locations [Fig 1(D)].

Multivariable analysis was performed with the inclusion of variables with *P*-value < 0.05 in the univariable analysis. As the intercorrelation between variables (tumor stage and clinical situation at the time of diagnosis of SpVT) was high, multicollinearity should be considered in the multivariable analysis using Cox proportional hazards models. Therefore, the two variables (tumor stage and clinical situation at the diagnosis of SpVT) were not included in the multivariable analysis at the same time. In multivariable analysis including age, ECOG PS, SpVT-related symptoms, and clinical situation at the time of diagnosis of SpVT (Table 3), ECOG PS, and clinical situation at the time of diagnosis of SpVT were independent significant prognostic factors. In the additional multivariable analysis including the variable tumor stage instead of clinical situation at the time of diagnosis of SpVT showed that ECOG PS and tumor stage were significant prognostic factors (S1 Table). Moreover, considering the period when SpVT can

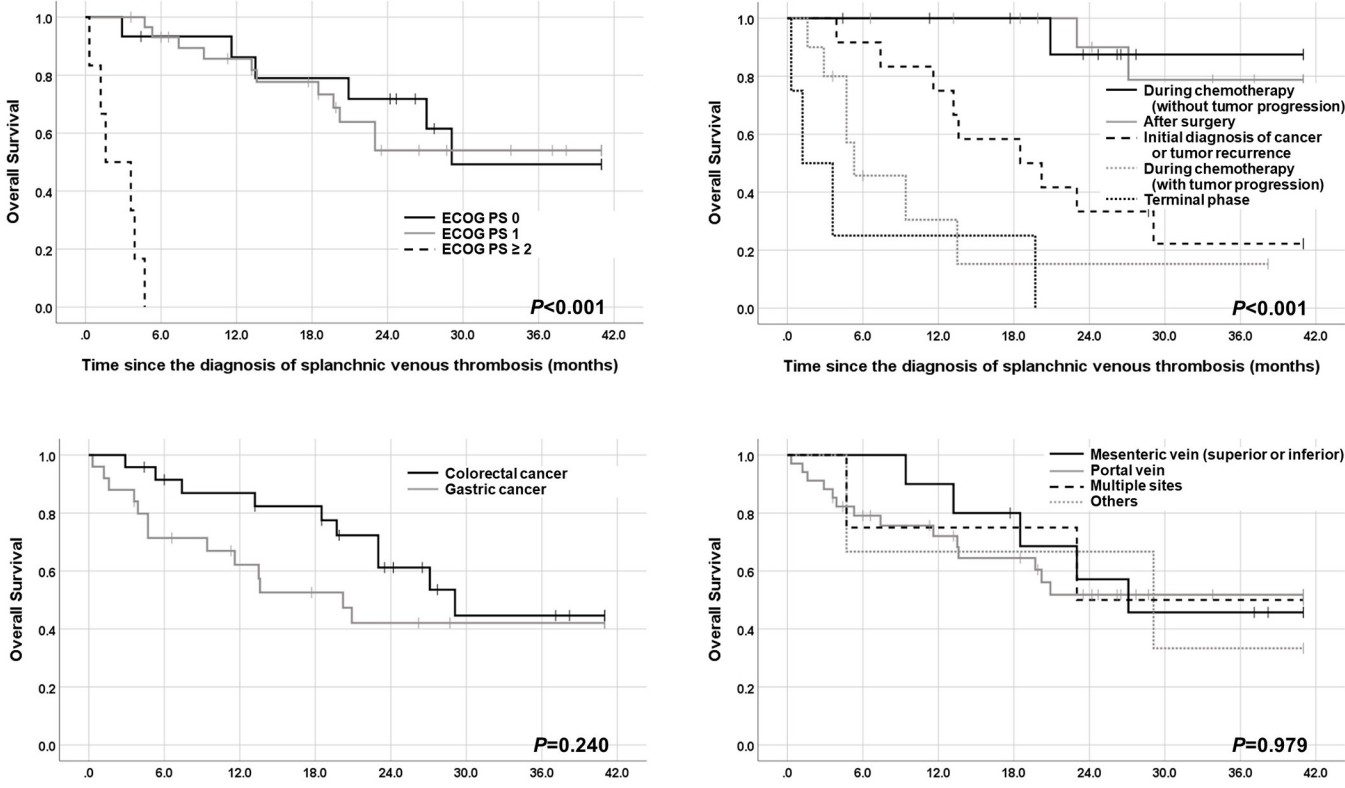

**Fig 1.** Survival analyses: (A) Kaplan–Meier curves of all patients (n = 51) comparing overall survival according to the Eastern Cooperative Oncology Group performance status; (B) Kaplan–Meier survival curves among patients (n = 51) with various clinical situations of developing splanchnic vein thrombosis; (C) Kaplan–Meier survival curves between patients with gastric cancer and those with colorectal cancer (n = 49); (D) Kaplan–Meier survival curves (n = 51) according to location of splanchnic vein thrombosis.

have a significant impact on survival, another survival analysis was repeated with a shorter follow-up period (12 months). In univariable analyses (S2 Table), ECOG PS, tumor stage, SpVT-related symptoms, clinical situation at the time of diagnosis of SpVT, albumin level, and hemoglobin level were associated with OS after the diagnosis of SpVT ($P < 0.05$). In multivariable analysis (S2 Table), ECOG PS, clinical situation at the time of diagnosis of SpVT, and albumin level were independently significant prognostic factors.

## Discussion

In this prospective study, the clinical characteristics and courses of SpVT were analyzed in patients with gastrointestinal cancer. SpVT occurred in various clinical situations and most of them did not cause symptoms or had little impact on the patient's cancer treatment course. During the follow-up period, in our patient cohort, there were no cases of death due to SpVT, and all deaths were caused by tumor progression. In the multivariable survival analysis, ECOG PS and clinical situation at the time of SpVT diagnosis were significant prognostic factors. To the best of our knowledge, this is the first prospective study of SpVT in patients with gastrointestinal cancer.

Recommendations on the treatment of SpVT in various guidelines are somewhat confusing and ambiguous [8]. According to the National Comprehensive Cancer Network guidelines [3], all patients with acute SpVT (symptoms/signs ≤ 8 weeks) are recommended to receive anticoagulation if no contraindication to anticoagulation exists. In cases with chronic SpVT (symptoms > 8 weeks, cavernous transformation/collateral noted, or signs of portal hypertension) or incidentally found SpVT, this guideline suggests weighing the risks and benefits of anticoagulation therapy on an individual basis. In the American Society of Clinical Oncology guideline [2], recommendations for the treatment of SpVT are not described in detail. This guideline only suggests that treatment of incidentally found SpVT should be offered on a case-by-case basis, considering the potential benefits and risks of anticoagulation. According to the American College of Chest Physicians guideline, which includes thrombosis that occurs in patients with or without cancer [9], anticoagulation is recommended over no anticoagulation in patients with symptomatic SpVT. In contrast, in patients with incidentally detected SpVT, anticoagulation is not recommended. However, even in the case of incidental SpVT, the American College of Chest Physicians guideline suggests that anticoagulation should be performed in the SpVT cases with the following factors: extensive thrombosis that appears to be acute (e.g., not present on a previous imaging study, presence of an intraluminal filling defect, lack of cavernous transformation), progression of thrombosis on follow-up imaging study, and ongoing cancer chemotherapy [9]. According to the American Association for the Study of Liver Diseases guideline [10], anticoagulation is recommended for all patients with acute portal vein thrombosis, including asymptomatic patients. In patients with chronic portal vein thrombosis, screening for gastroesophageal varices is recommended in all patients. Long-term anticoagulation therapy is suggested in patients without cirrhosis, and with a permanent risk factor for venous thrombosis that cannot be corrected otherwise, provided no major contraindication exists. However, the American Association for the Study of Liver Diseases guideline was written for patients with liver disease and was not intended primarily for cancer patients.

In our study, we found that most SpVT occurred in the blood vessels adjacent to the existing tumor, except in the SpVT cases that developed during the postoperative or adjuvant chemotherapy periods. For example, we could not detect inferior mesenteric vein thrombosis in patients with gastric cancer. This phenomenon can be explained by the previously suggested mechanisms of thrombosis in cancer. According to the literature [11–13], tumors can compress veins, resulting in venous stasis, thus leading to thrombosis. Moreover, tumor cells

release various substances that promote thromboembolic events, such as tissue factor, micro-particles, and cancer procoagulants. Therefore, blood vessels closer to tumor cells are more likely to be affected by these substances. This phenomenon is considered to have important significance in the therapeutic aspects of SpVT. In cases where the intra-abdominal tumor cannot be removed, for example, in patients undergoing palliative chemotherapy or end-of-life care, the tumor itself is the cause of the majority of SpVT cases. Therefore, even if anticoagulation is performed, the effect is limited, and temporary anticoagulation is unlikely to be helpful. Given the risk-benefit considerations, careful judgement should be made as to whether to administer long-term anticoagulation to these patients.

Interestingly, we found that SpVT was incidentally diagnosed and did not cause clinical problems in most patients with gastrointestinal cancers. Most cases of SpVT that developed during postoperative period or chemotherapy without tumor progression (including adjuvant chemotherapy) improved spontaneously without anticoagulant treatment. Even in SpVT cases that developed during palliative chemotherapy or end-of-life care, most cases of SpVT remained stable. Although some SpVT cases (31.4%) showed extension within the same vein or to adjacent other veins, most of them did not cause symptoms or had little effect on the patient's cancer treatment journey. Moreover, when the SpVT occurred in the end-of-life care period, the patients only survived for a short time (median OS, 1.2 months); thus, anticoagulation in these patients would not be justified. Importantly, no patient died from SpVT in our patient cohort. At the time of data cutoff, 23 patients died, and all died of tumor progression. ECOG PS and clinical situation at the time of diagnosis of SpVT were independent prognostic factors in both short-term and long-term survival analyses. In short-term survival analysis (12 months), albumin level was also an independent prognostic factor, which is consistent with the results of previous studies in cancer patients [14,15].

Our observations are generally in line with the recommendations of the current guidelines [2,3,9,10]. In our patient cohort, most incidentally detected acute or chronic SpVT did not require anticoagulation. Therefore, we strongly suggest that, even in patients with active cancer who do not have SpVT-related symptoms, doctors carefully monitor symptoms without anticoagulation regardless of whether SpVT is acute or not. Although the patient group was different from our study, some authors also suggested a watchful waiting approach for isolated portal or splenic vein thrombosis in patients with liver cirrhosis, based on a retrospective study showing spontaneous thrombus regression in 47% of patients, stability in 45%, and progression in only 7% [16]. However, in cases with symptomatic SpVT, we fully agree with the use of anticoagulants, as in the current guidelines.

With regard to treatment of SpVT, although the direct evidence is limited, anticoagulation with low molecular weight heparin (LMWH) or direct oral anticoagulant could be considered for patients with symptomatic SpVT. A previous systematic review and meta-analysis demonstrated that anticoagulant therapy improves SpVT recanalization and reduces the risk of thrombosis progression without increasing major bleeding [17]. LMWH may be preferred, particularly for patients with upper gastrointestinal cancer, because studies have shown that patients with upper gastrointestinal cancer tend to experience more bleeding complications with direct oral anticoagulant than with LMWH [18–20]. The optimal duration of anticoagulant therapy has not yet been established [21]. In cases with incidentally detected or chronic SpVT, as mentioned above, careful observation and symptom monitoring would be preferred to prompt anticoagulation. In our experience, even most cases with progression in the extent of SpVT did not require anticoagulation because progressive SpVT did not cause symptoms or had little impact on the patient's status or survival outcome.

This study has several limitations. First, only patients of Korean ethnicity were included. Further studies are needed to confirm our observation because ethnicity is well known to

significantly affect the incidence of VTE [6,7,22–27]. Second, gastric and colorectal cancers were mainly analyzed, and other solid tumors were not included in this study. Third, the number of patients enrolled in this study was not large although there was no problem in drawing conclusions. We conducted post hoc power analysis. In this study, 9 out of 51 patients (9/51 = 0.176 [$P_1$]) received anticoagulant treatment. When it was assumed that about one-third of patients with SpVT will require anticoagulation ($P_0$ = 0.35 [null hypothesis]), the sample size of 51 achieved 73.379% power to detect a difference ($P_1$-$P_0$) of -0.174 with a significance level (alpha) of 0.05. When it was assumed that about half of patients with SpVT will require anticoagulation ($P_0$ = 0.50), the sample size of 51 achieved 99.955% power to detect a difference ($P_1$-$P_0$) of -0.324 (alpha = 0.05; see S1 Appendix for post hoc power analysis). Finally, in some cases with abdominal symptoms, such as abdominal discomfort, pain or ascites, distinguishing whether these symptoms were caused by tumors or SpVT was difficult. For example, in patients with portal vein thrombosis and peritoneal metastasis, distinguishing whether the cause of ascites was peritoneal metastasis or portal vein thrombosis was not easy when cancer cells were negative in cytology.

## Conclusions

In patients with gastrointestinal cancer, most SpVT cases are asymptomatic and incidentally found. Most cases of SpVT had little effect on the patient's cancer treatment journey. Therefore, except for symptomatic SpVT cases, we suggest that clinicians could adopt a strategy of close observation and symptom monitoring regardless of whether SpVT is acute or not, even in patients with active cancer. Anticoagulation should be considered only for SpVT cases selected strictly, weighing the risks and benefits.

## Supporting information

**S1 Fig. Kaplan–Meier curves for overall survival in all patients (n = 51).**
(JPG)

**S2 Fig. Kaplan–Meier curves comparing overall survival (n = 51) according to tumor stage.**
(JPG)

**S1 Table. Univariable and multivariable analyses on prognostic factors in patients with splanchnic vein thrombosis.**
(DOCX)

**S2 Table. Univariable and multivariable analyses on prognostic factors in patients with splanchnic vein thrombosis (repeated survival analysis with a short time of follow-up, 12 months).**
(DOCX)

**S1 Appendix. Post hoc power analysis.**
(DOCX)

**S1 File. Minimal data set.**
(XLSX)

## Acknowledgments

The authors thank all patients who participated in this study. We are grateful to Medical Research Collaborating Center (MRCC) at Seoul National University Bundang Hospital for the assistance in the revision of this manuscript.

## Author Contributions

**Conceptualization:** Ja Min Byun, Keun-Wook Lee.

**Data curation:** Minsu Kang, Keun-Wook Lee.

**Formal analysis:** Minsu Kang, Keun-Wook Lee.

**Funding acquisition:** Keun-Wook Lee.

**Investigation:** Minsu Kang, Koung Jin Suh, Ji-Won Kim, Jin Won Kim, Keun-Wook Lee.

**Methodology:** Minsu Kang, Ja Min Byun, Keun-Wook Lee.

**Project administration:** Keun-Wook Lee.

**Resources:** Koung Jin Suh, Ji-Won Kim, Jin Won Kim, Ji Yun Lee, Jeong-Ok Lee, Soo-Mee Bang, Yu Jung Kim, Se Hyun Kim, Jee Hyun Kim, Jong Seok Lee, Keun-Wook Lee.

**Validation:** Minsu Kang, Keun-Wook Lee.

**Visualization:** Minsu Kang, Keun-Wook Lee.

**Writing – original draft:** Minsu Kang, Keun-Wook Lee.

**Writing – review & editing:** Minsu Kang, Koung Jin Suh, Ji-Won Kim, Ja Min Byun, Jin Won Kim, Ji Yun Lee, Jeong-Ok Lee, Soo-Mee Bang, Yu Jung Kim, Se Hyun Kim, Jee Hyun Kim, Jong Seok Lee, Keun-Wook Lee.

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
