## [Decision Letter · Decision Letter 0]

30 Jun 2021

PONE-D-21-10010

Clinical characteristics and disease course of splanchnic vein thrombosis in gastrointestinal cancers: A prospective cohort study

PLOS ONE

Dear Dr. Lee,

Thank you for submitting your manuscript to PLOS ONE. After careful consideration, we feel that it has merit but does not fully meet PLOS ONE’s publication criteria as it currently stands. Therefore, we invite you to submit a revised version of the manuscript that addresses the points raised during the review process.

Splanchnic vein thrombosis is a serious condition especially in GI cancers. The authors have provide a clinical description of a cohort and try to identify risk factors for mortality.

However, the reviewers raise some issues that needs to be addressed.

In addition to these, please provide any information, in available, on presence of other thrombophilia factors,

Also, please proof edit carefully or use an editing service such as AJE.

We look forward to receiving your revised manuscript.

Kind regards,

Pal Bela Szecsi, M.D. D.M.Sci.

Academic Editor

PLOS ONE

Journal Requirements:

2. Please provide a sample size and power calculation in the Methods, or discuss the reasons for not performing one before study initiation

"This research was partially funded by the Seoul National University Bundang Hospital

Research Fund (Number: 02-2017-036). The funders had no role in study design, data

collection and analysis, decision to publish, or preparation of the manuscript."

Reviewers' comments:

Reviewer's Responses to Questions

**Comments to the Author**

1. Is the manuscript technically sound, and do the data support the conclusions?

Reviewer #1: Yes

Reviewer #2: Partly

2. Has the statistical analysis been performed appropriately and rigorously? 

Reviewer #1: Yes

Reviewer #2: Yes

3. Have the authors made all data underlying the findings in their manuscript fully available?

Reviewer #1: Yes

Reviewer #2: Yes

4. Is the manuscript presented in an intelligible fashion and written in standard English?

Reviewer #1: Yes

Reviewer #2: No

5. Review Comments to the Author

Reviewer #1: The study involves a prospective analysis of a small cohort of subjects with splanchnic vein thrombosis, mainly in gastric and colorectal cancers. Although limited by its size, it is technically sound with appropriate statistical analysis and presented in an easily comprehensible manner.

I would request the authors to clarify a couple things-

1.Line 148, Page 9- What does 'aggravation' of SpVT mean? Does it mean extension of SpVT or development of new SpVT related symptoms or both?

2. Line 146, Page 10- Please clarify what lead to the 4 patients being treated with anticoagulation upon initial diagnosis? Were they all symptomatic or had concomitant DVT/PE or any other factors?

3. Did any patients in the study suffer from SpVT related complications such as mesenteric ischemia or portal hypertension?

Reviewer #2: Major comments:

1. In the "Materials and methods" section please specify:

- if patients is consecutively enrolled

- the outcome of interest. Authors reported in the Results mortality rate, recurrent venous thrombotic events, and the rate of vessel recanalization without any information in the Material and methods section. Furthermore, authors should report (if available) major and clinically relevant non-major bleeding occurred during follow-up.

-how the diagnosis of Splanchnic vein thrombosis is performed and how patients with splanchnic vein thrombosis is managed (e.g. anticoagulant regimens administered).

-duration of follow-up and how follow-up is performed (e.g. ultrasonography or computed tomography for vein recanalization or thrombosis progression).

Furthermore, authors should better explain the statistical methods, how they chose a priori the variables included in the model, and any sensitivity analysis performed.

2. Results: authors should consider to repeat survival analysis with a short time of follow-up (e.g., 6 or 12 months). Patients with cancer and splanchnic vein thrombosis generally have a short prognosis and after the first year of follow-up mortality may be affected by other variables.

3. Discussion: please report in first paragraph the major findings of the study.

Please consider to shorten the Discussion as some parts appear confusing and ripetitive.

Minor comments:

-Please report the number of patients lost to follow-up in the first part of Results section

-Please be consistent with the use of "gastric cancer" in the text and tables

-Please replace "symptomatic" or "incidental" thrombosis with "incidentally-detected thrombosis"

-on row 114 page 7, please specify the type of other thrombosis

6. PLOS authors have the option to publish the peer review history of their article (what does this mean?). If published, this will include your full peer review and any attached files.

Reviewer #1: No

Reviewer #2: No

---

## [Author Response · Author response to Decision Letter 0]

16 Aug 2021

Please refer to the uploaded 'Response to Reviewers.docx' file. 

Thank you.

---

## [Decision Letter · Decision Letter 1]

7 Oct 2021

PONE-D-21-10010R1Clinical characteristics and disease course of splanchnic vein thrombosis in gastrointestinal cancers: A prospective cohort studyPLOS ONE

Dear Dr. Lee,

Thank you for submitting your manuscript to PLOS ONE. After careful consideration, we feel that it has merit but does not fully meet PLOS ONE’s publication criteria as it currently stands. Therefore, we invite you to submit a revised version of the manuscript that addresses the points raised during the review process.

ACADEMIC EDITOR:

The paper has improved considerably, however one of the reviewer still have issues that needs to be addressed.

I understand that a power calculation was not performed beforehand. Please perfome one post-hoc? 

We look forward to receiving your revised manuscript.

Kind regards,

Pal Bela Szecsi, M.D. D.M.Sci.

Academic Editor

PLOS ONE

Reviewers' comments:

Reviewer's Responses to Questions

**Comments to the Author**

1. If the authors have adequately addressed your comments raised in a previous round of review and you feel that this manuscript is now acceptable for publication, you may indicate that here to bypass the “Comments to the Author” section, enter your conflict of interest statement in the “Confidential to Editor” section, and submit your "Accept" recommendation.

Reviewer #1: All comments have been addressed

Reviewer #2: (No Response)

2. Is the manuscript technically sound, and do the data support the conclusions?

Reviewer #1: Yes

Reviewer #2: Partly

3. Has the statistical analysis been performed appropriately and rigorously? 

Reviewer #1: Yes

Reviewer #2: Yes

4. Have the authors made all data underlying the findings in their manuscript fully available?

Reviewer #1: Yes

Reviewer #2: Yes

5. Is the manuscript presented in an intelligible fashion and written in standard English?

Reviewer #1: Yes

Reviewer #2: Yes

6. Review Comments to the Author

Reviewer #1: The study makes a small yet important contribution to the field of GI malignancies related splanchnic vein thrombosis, which has so far been under-researched. It is presented in an easily comprehensible manner with sound statistical analysis and comprehensive data collection of important variables in the study cohort.

Reviewer #2: Major revision

Methods

1. Please evaluate to better specify the outcomes evaluated. In Results both recurrent event and bleeding were reported without any clarification in the Methods. Furthermore, authors should describe as outcome were defined and evaluated.

2. Authors should clearer report statistical methods used and the sensitivity analysis performed. Furthermore, they should report in methods the univariable and multivariable models actually reported in the Results. Furthermore, they should report a-priori the variable they evaluate in the model.

Results

Author should evaluate:

- to report time from surgery, chemotherapy or radiotherapy from thrombosis development (e.g., < or > 3 months), if available

- to report in the text the number of patient with different type of thrombosis or with different rate of recanalization along with proportions. This may increase the readability of results.

- to report more detailed information on anticoagulant therapy (e.g. number of patients treated with different type of anticoagulant therapy, duration of anticoagulation).

- to report the number of outcomes events in both anticoagulated patients and those that remained untreated.

Discussion and conclusion

Discussion may be shortened to increase readability.

Conclusion should better report the results of the study (e.g. reported data do not permit to draft any conclusion on anticoagulant therapy).

Minor revision

Author should correct typo and grammatical errors.

Author should consider include in introduction and discussion recent data from prospective studies on clinical course of cancer patients with splanchnic vein thrombosis.

7. PLOS authors have the option to publish the peer review history of their article (what does this mean?). If published, this will include your full peer review and any attached files.

Reviewer #1: No

Reviewer #2: No

---

## [Author Response · Author response to Decision Letter 1]

25 Oct 2021

Please find the attached file 'Response to Reviewers.docx'

---

## [Decision Letter · Decision Letter 2]

29 Nov 2021

PONE-D-21-10010R2Clinical characteristics and disease course of splanchnic vein thrombosis in gastrointestinal cancers: A prospective cohort studyPLOS ONE

Dear Dr. Lee,

Thank you for submitting your manuscript to PLOS ONE. After careful consideration, we feel that it has merit but does not fully meet PLOS ONE’s publication criteria as it currently stands. Therefore, we invite you to submit a revised version of the manuscript that addresses the points raised during the review process.

Both reviewers are satisfied, however I request that you include a paragraph regarding the power calculation you performed. It strength the statement of use of anticoagulation, especially considering the relative few patients in the study.

We look forward to receiving your revised manuscript.

Kind regards,

Pal Bela Szecsi, M.D. D.M.Sci.

Academic Editor

PLOS ONE

Journal Requirements:

Reviewers' comments:

Reviewer's Responses to Questions

**Comments to the Author**

1. If the authors have adequately addressed your comments raised in a previous round of review and you feel that this manuscript is now acceptable for publication, you may indicate that here to bypass the “Comments to the Author” section, enter your conflict of interest statement in the “Confidential to Editor” section, and submit your "Accept" recommendation.

Reviewer #1: All comments have been addressed

Reviewer #2: All comments have been addressed

2. Is the manuscript technically sound, and do the data support the conclusions?

Reviewer #1: Yes

Reviewer #2: Yes

3. Has the statistical analysis been performed appropriately and rigorously? 

Reviewer #1: Yes

Reviewer #2: Yes

4. Have the authors made all data underlying the findings in their manuscript fully available?

Reviewer #1: Yes

Reviewer #2: Yes

5. Is the manuscript presented in an intelligible fashion and written in standard English?

Reviewer #1: Yes

Reviewer #2: Yes

6. Review Comments to the Author

Reviewer #1: (No Response)

Reviewer #2: (No Response)

7. PLOS authors have the option to publish the peer review history of their article (what does this mean?). If published, this will include your full peer review and any attached files.

Reviewer #1: No

Reviewer #2: No

---

## [Author Response · Author response to Decision Letter 2]

6 Dec 2021

Please see the uploaded 'Response to Reviewers.docx' file.

---

## [Editor Report · Decision Letter 3]

9 Dec 2021

Clinical characteristics and disease course of splanchnic vein thrombosis in gastrointestinal cancers: A prospective cohort study

PONE-D-21-10010R3

Dear Dr. Lee,

We’re pleased to inform you that your manuscript has been judged scientifically suitable for publication and will be formally accepted for publication once it meets all outstanding technical requirements.

Kind regards,

Pal Bela Szecsi, M.D. D.M.Sci.

Academic Editor

PLOS ONE

---

## [Editor Report · Acceptance letter]

20 Dec 2021

PONE-D-21-10010R3 

Clinical characteristics and disease course of splanchnic vein thrombosis in gastrointestinal cancers: A prospective cohort study 

Dear Dr. Lee:

I'm pleased to inform you that your manuscript has been deemed suitable for publication in PLOS ONE. Congratulations! Your manuscript is now with our production department. 

Kind regards, 

on behalf of

Dr. Pal Bela Szecsi 

Academic Editor

PLOS ONE